# Microelectromechanical System Measurement of Platelet Contraction: Direct Interrogation of Myosin Light Chain Phosphorylation

**DOI:** 10.3390/ijms22126448

**Published:** 2021-06-16

**Authors:** Mitchell J. George, Julia Litvinov, Kevin Aroom, Leland J. Spangler, Henry Caplan, Charles E. Wade, Charles S. Cox, Brijesh S. Gill

**Affiliations:** 1Department of Surgery, McGovern Medical School at The University of Texas Health Science Center, Houston, TX 77030, USA; kevin.r.aroom@uth.tmc.edu (K.A.); charles.e.wade@uth.tmc.edu (C.E.W.); charles.s.cox@uth.tmc.edu (C.S.C.J.); brijesh.s.gill@uth.tmc.edu (B.S.G.); 2Department of Pediatric Surgery, McGovern Medical School at The University of Texas Health Science Center, Houston, TX 77030, USA; julia.litvinov@uth.tmc.edu (J.L.); henry.w.caplan@uth.tmc.edu (H.C.); 3Aspen Microsystems, LLC, Manitou Springs, CO 80829, USA; chip@aspenmicrosystem.com

**Keywords:** platelets, myosin light chain, phosphorylation, force of contraction, Western blot

## Abstract

Myosin Light Chain (MLC) regulates platelet contraction through its phosphorylation by Myosin Light Chain Kinase (MLCK) or dephosphorylation by Myosin Light Chain Phosphatase (MLCP). The correlation between platelet contraction force and levels of MLC phosphorylation is unknown. We investigate the relationship between platelet contraction force and MLC phosphorylation using a novel microelectromechanical (MEMS) based clot contraction sensor (CCS). The MLCK and MLCP pair were interrogated by inhibitors and activators of platelet function. The CCS was fabricated from silicon using photolithography techniques and force was validated over a range of deflection for different chip spring constants. The force of platelet contraction measured by the clot contraction sensor (CCS) was compared to the degree of MLC phosphorylation by Western Blotting (WB) and ELISA. Stimulators of MLC phosphorylation produced higher contraction force, higher phosphorylated MLC signal in ELISA and higher intensity bands in WB. Inhibitors of MLC phosphorylation produced the opposite. Contraction force is linearly related to levels of phosphorylated MLC. Direct measurements of clot contractile force are possible using a MEMS sensor platform and correlate linearly with the degree of MLC phosphorylation during coagulation. Measured force represents the mechanical output of the actin/myosin motor in platelets regulated by myosin light chain phosphorylation.

## 1. Introduction

Blood clots are active structures formed from a composite of fibrin, a structural polymer, and platelet aggregates which serve as the composite adhesive that binds individual fibrin strands [1]. Coagulation begins when individual platelets aggregate at a site of endothelial injury and then induce the polymerization of fibrinogen, the soluble circulating monomer of fibrin. Platelets deliver diverse molecular payloads that drive the coagulation system from a normal quiescent state of hemostasis into the positive-feedback-driven cascades of thrombosis. The emerging clot structure binds to exposed collagen and then the platelets contract, greatly enhancing the structural strength of the clot and allowing it to oppose the pressure of blood [2].

Platelet contraction is driven by the cycling of myosin IIA upon actin filaments. Force generation in platelets is determined by phosphorylation of a regulatory domain in the myosin light chain which is controlled by a myosin light chain kinase (MLCK) and myosin light chain phosphatase (MLCP) pair similar to that found in smooth muscle [3,4,5]. Platelet signaling cascades converge onto the MLCK/MLCP pair as seen in Figure 1 [6]. This proposed diagram describes the feedback mechanisms that control platelet contraction and various inhibitors or activators of this system.

Platelets provide most of the strength of a blood clot by contracting along fibrin strands [7]. Contraction of the human blood clot was first described scientifically by William Hewson [8]. Early attempts to quantify clot contraction and plasma syneresis were performed by MacFarlane [9]. Modern experiments to measure directly the force produced in bulk clot contraction were first successfully undertaken by Carr et al. utilizing an isometric strain gage method [10]. Force developed by individual platelets has been determined more recently via atomic force microscopy; a single platelet generates on average a maximum contractile force of 29 nN [2,11]. Platelets utilize a version of the actin/myosin molecular motor to generate force and were the first non-muscle cell in which this motor was discovered [12].

Platelet contraction is clinically important and presents a potentially attractive diagnostic target for a number of reasons. Platelets are accessible—blood samples may readily be obtained from peripheral venipuncture. Platelet contraction requires intact aggregation, adhesion, and paracrine signaling and thus represents the final common pathway of platelet activation. Platelets, with no nuclei, may utilize somewhat less complex signal transduction cascades than nucleated cells [13].

This study aims to prove the relationship between myosin light chain (MLC) phosphorylation and platelet contraction force via inhibitors or activators of the MLCK/MLCP pair in a novel silicon wafer-based force assay. This assay is different from clot contraction assays described by Tutwiler et al. in that it measures the force of platelet contraction directly [14]. Because MLC signaling is a highly conserved system, platelet MLC signaling is expected to reflect that seen in smooth muscle. However, multiple platelet-specific isoforms exist within the upstream MLC signal transduction cascade and platelet MLC signaling has emerged on a piecemeal basis [4,15,16]. We provide confirmatory and new data on platelet contraction force and MLC phosphorylation at multiple points throughout this cascade.

## 2. Results

### 2.1. Linearity of MEMS Clot Contraction Sensor

Linearity ranged from 96% to 99% for all three spring constants within the 0–100 micrometer deflection range (Figure 2A). The 100 N/m spring constant offers the optimal spectrum of force detection. It remains linear within the maximum anticipated force range and allows better resolution than the 300 N/m chip. The 5 N/m chip is unable to detect a high enough force within the desired deflection range and is subject to noise from small vibrations transmitted from the environment in addition to being very fragile. Platelet contraction force over a standard 3600 s assay is shown in Figure 2B.

### 2.2. Force of Platelet Contraction Measurements

Normal contraction force with only calcium restoration and no reagent added was 2958 ± 373 microNewtons (n = 5, Figure 3A). If calcium is not added to the sample, no force is generated since no clot forms (data not shown). U46619 is the synthetic analog of thromboxane A2 and stimulates the highest force observed for all subjects compared to controls. The average max force for U46619 was 7337 ± 1408 microNewtons which was 60% higher compared to controls (Figure 4A). The percent force increase due to the other activators PMA and Okadaic Acid was 41.6% and 46.8% respectively.

Cpd7a is a quinoline derivative that inhibits PDE5 and demonstrates the greatest inhibition of platelet contraction force. The average max force for Cpd7a was 922 ± 504 microNewtons which was 69% lower compared to controls. The percent force decrease due to the addition of other inhibitors were 53.5% for ML-7 and MLCK18, 63% for K252d, and 65% for Y27632 compared to non-inhibited controls.

Sildenafil is a well-known inhibitor of PDE5 that paradoxically increases the force of platelet contraction (34.6%). Combination of sildenafil with Y27632 results in a diminished force, suggesting the involvement of the ROCK pathway in the action of sildenafil on platelet contraction (data not shown).

### 2.3. ELISA on MLC Phosphorylation

Trends of MLC phosphorylation mirrored the activation or inhibition of platelet contraction using the same array of previously described reagents (Figure 3B). The increase in expression of pMLC relative to controls due to the addition of activators sildenafil, U46619, PMA, and okadaic acid was 40.9%, 41.3%, 30.5%, and 30.7%, respectively. The decrease in pMLC expression due to addition of inhibitors was 38.5% for Cpd7a, 42.8% for ML-7, 39.8% for K252d, and 46.0% for Y27632. Since the ELISA was performed in-cell, all stimulation or inhibition of the platelets were conducted on live platelets within 24 h of the blood draw to ensure platelet viability and function.

### 2.4. Correlation of Platelet Contraction Force to MLC Phosphorylation

MLC phosphorylation correlates linearly with platelet contraction force with an R^2^ value of 0.91 (Figure 4). For each point, the *x*-axis value is defined by the average of the platelet contraction force of the 5 volunteers. The *y*-axis values are averaged from levels of phosphorylated MLC.

### 2.5. Western Blotting

After platelet stimulation, protein lysates were extracted and WB was performed to detect levels of pMLC (Figure 5). Reagents that were known to increase the platelet force of contraction demonstrated an increase in band intensity of pMLC (U46619, sildenafil, PMA, okadaic acid). Conversely, reagents known to decrease the force of platelet contraction showed a decrease in band intensity (ML-7, Cpd7a, K252d, Y27632).

## 3. Discussion

Our results demonstrate that platelet contractile force can be reliably measured by a MEMS force transducer and is governed by the dynamics of MLC phosphorylation in platelets. We probed the signal transduction pathways controlling the MLCK/MLCP pair with activators and inhibitors using a silicon wafer-based MEMS device to measure the force generated by clot contraction. Changes in platelet pMLC after physiologic or artificial stimulation correlate linearly with changes in platelet contraction forces. Reagents that increase levels of pMLC increase the force of platelet contraction and vice versa. Levels of pMLC were measured using ELISA and corroborated with Western Blotting experiments.

A number of reagents act to increase levels of pMLC as measured by ELISA and Western Blot. Okadaic acid is a cell-permeable polycyclic ether that induces platelet shape change and increases phosphorylation on MLC via inhibition of MLC phosphatase [15]. Its known increase of MLC phosphorylation and its predicted effect of increased platelet contraction force was confirmed in this study. PMA is a synthetic analog to diacylglycerol and induces phosphorylation of a 47-kDa substrate of protein kinase C, which leads to platelet aggregation, the release of dense granule content, and a rise in cytoplasmic Ca [16]. Similarly, the addition of PMA also increased the force of contraction. U46619 is a MEK/ERK pathway activator and analog of thromboxane A2 which acts through a thromboxane receptor [17] and it produced the highest increase in MLC phosphorylation and platelet contraction force. These activators act in widely different manners but result in the same downstream measurable effect of increased platelet contraction force.

Sildenafil is a PDE5 inhibitor [18] and unexpectedly leads to higher levels of pMLC and higher platelet contraction forces. The established mechanism of sildenafil is to inhibit PDE5 which leads to increased levels of cGMP, increased levels of PKG, decreased levels of PKC, and ultimately increased levels of MLCP and thus expected decreased levels of pMLC. One potential explanation for this discrepancy in expected platelet contraction force is decreased levels of fibrinogen. In a rabbit model, El-Sayed et al. demonstrated administration of sildenafil led to decreased levels of fibrinogen [19]. Soluble fibrinogen converts into a solid fibrin gel during the clotting process and offers rigidity and strength to a forming clot. Thus, the clot contraction sensor would detect more displacement and higher contraction forces in samples with lower fibrinogen levels. An explanation for the discrepancy in expected levels of pMLC is a disassociation of force from levels of pMLC. Chuang et al. demonstrated in rabbit smooth muscle cells higher than expected levels of pMLC in the setting of sildenafil induced relaxation [20]. These findings suggest a possible new pathway that uncouples levels of pMLC from actin and myosin cycling in platelets. Lastly, platelets activated with Sildenafil secrete ATP which causes secondary platelet activation and higher platelet contraction forces [21,22].

ML-7 directly inhibits the catalytic activity of MLCK [23]. Y27632 is a selective p160 ROCK inhibitor, which competes with ATP for binding to the catalytic site of p160 and regulates smooth muscle contraction [24,25,26]. In platelets, this results in a decrease of GPIIb/IIIa activation [27], amount of ADP required to induce platelet shape change and a decrease in phosphorylation of myosin light chain [28]. K252d inhibitor is an indolocarbazole alkaloid found in some soil bacteria and serves as a mild PKC inhibitor and a Ca/Calmodulin-dependent phosphodiesterase inhibitor [29]. The mechanism of action of this compound is not known in platelets. Since baseline platelet pMLC can be considered somewhat irreversible, the degree of decrease in pMLC will generally be small as measured by ELISA.

The findings in this manuscript are clinically significant because we describe reliable methods to control and analyze platelet function through modulation of its governing MLCP/MLCK pair. Other established methods of measuring platelet function like aggregometry or adhesion do not have well-established signaling cascades with a single controlling kinase/phosphatase pair like a contraction. In addition, measuring contraction force offers a universally understood unit of measure for clinical comparison. For this reason, platelet contraction is a unique and advantageous phase of platelet function to measure. In addition, the MLCP/MLCK pair presents an attractive target for possible future pharmacotherapy once the clinical significance of platelet contraction is better understood. Certain studies like our previous work in severely injured trauma patients have begun to describe the clinical significance of platelet contraction [30]. However, more clinical studies in expanded patient subsets like cardiac or renal disease are needed to better define the clinical meaning of platelet contraction.

## 4. Materials and Methods

The Materials and Methods should be described with sufficient details to allow others to replicate and build on the published results. Please note that the publication of your manuscript implicates that you must make all materials, data, computer code, and protocols associated with the publication available to readers. Please disclose at the submission stage any restrictions on the availability of materials or information. New methods and protocols should be described in detail while well-established methods can be briefly described and appropriately cited.

Research manuscripts reporting large datasets that are deposited in a publicly available database should specify where the data have been deposited and provide the relevant accession numbers. If the accession numbers have not yet been obtained at the time of submission, please state that they will be provided during review. They must be provided prior to publication.

Interventionary studies involving animals or humans, and other studies that require ethical approval, must list the authority that provided approval and the corresponding ethical approval code.

Design of the Clot Contraction Sensor: This iteration of a clot contraction sensor (CCS) is an improvement upon a previously published device that uses a Nickel wire cantilever to measure clot contraction [31]. The new MEMS-based clot contraction sensor (CCS) described here measures platelet contraction force in units of micro-Newtons. The CCS (Figure 6A) consists of a sensor platform attached to a sensor frame by sixteen torsion beam springs. The beam springs are configured in a symmetric arrangement that allows a linear translation of the sensor platform perpendicular to the plane of the sensor frame. Hooke’s Law defines the relationship between displacement (*x*) of the sensor platform from the sensor frame and the resultant force (*F*) generated by the beam springs with linear spring constant *k*:*F* = *kx*

We targeted the CCS design to have a maximum displacement of 100 microns with an applied force of 10,000 micro-Newtons. The torsion beams are arranged around the periphery of the sensor platform in such a manner that two torsion beams span from the sensor platform to a connection arm near the center of the platform. The other end of the connection arm is attached to the second set of torsion beams that span from the connection arm to the sensor frame. These four torsion beams nominally have the same length, width, and thickness. All four sides of the sensor platform have this arrangement of four torsion beams with a central connection arm leading to a total of 16 torsion beams in the CCS. The length of the arm between the two sets of torsion beams, along with the dimensions of the torsion beams yields the overall spring constant of the CCS. The arrangement of the torsion beams and connecting arms is such that a linear displacement occurs with the restoring force being created through torsional springs.

To calculate the linear displacement of the sensor platform that is suspended by the sixteen torsion beams, one begins with the angular form of Hooke’s law which defines the torque (τ) on a single beam as:τ = ĸθ
where ĸ is the torsion spring constant and θ is the angle of twist. The torsion spring constant ĸ for a beam is a function of the length of the beam, its cross-sectional geometry, and the shear modulus of the beam material. Silicon microfabrication technology can create beams with a uniform rectangular cross-section which result in a torsion spring constant of
ĸ = L/(GJ)
where L is the length of the torsion beam, G is the shear modulus of silicon and J is the torsional constant of a beam with a rectangular cross-section. J is approximated by:J = βab^3^

With β being a factor that is based on the moment of inertia of a rectangle, with (a) being the thickness of the beam and (b) being the width of the beam. For a CCS with torsion beams that are 10 microns wide and 50 microns thick, β = 0.291 [32]. The length of the connection arm that spans between the two sets of torsion beams is (C) and it is assumed to be perfectly stiff as are the anchors on the sensor platform and sensor frame. With these assumptions applied to the four torsion beam central “arm” configuration, the displacement (x) is:x = Csin(θ)

While there are an infinite combination of torsion beam length, thickness, and width, and connection arm lengths that meet the objective of a 100-micron displacement at 10,000 micro-Newton force, the desire to have a 4 mm square sensor platform that is 50 microns thick combined with other practicalities lead us to a torsion beam length and width of approximately 680 and 10 microns respectively. The CCS (Coagulex, Inc., Houston, TX, USA) is fabricated with a silicon wafer-based microelectromechanical system (MEMS) process. The devices are fabricated on a six-inch silicon-on-insulator substrate utilizing deep reactive ion etching (DRIE) to produce the mechanical features. Devices are designed to provide a nominal stroke of 100 microns and a spring constant of 100 N/m at the platform.

The CCS cartridge is assembled from seven components including a blood sample (Figure 6B). The cartridge resides in a heated chamber maintained at 37 degrees Celsius (not shown). A study sample of citrated and recalcified blood (7) is injected between two 6 mm diameter acrylic plates (1&6) and allowed to clot. The plates are 1 mm apart and the study sample is 35 micro-liters. As the study sample clots, the platelets within the sample contract and translate the superior acrylic plate (6) downward. The superior acrylic plate is attached to the sensor platform of the CCS (5). The CCS sensor platform (5) is attached to the sensor frame (3) by four beam springs. The sensor frame (3) of the CCS is fixed to the device platform (2). A camera with a microscope objective focuses on a tracking prism (4) to determine the downward displacement of the superior acrylic plate in micrometers. An orthogonal view of the entire device assembly is shown in Figure 6C and a cross-sectional view through the CCS cartridge is shown in Figure 6D.

A LabVIEW (National Instruments, Austin, TX, USA) software program calculates force from the recorded displacement of the CCS sensor platform. The device signal reflects the contraction force of platelets within the forming clot of the blood sample. The assay length is one hour.

Linearity of MEMS Clot Contraction Sensor: System performance is determined by the linearity of the MEMS sensor within a specified deflection range of 100 micrometers. This deflection range is established by the field of view of the camera watching the chip during a testing cycle. Chips of differing spring constant were tested to determine the optimal range of force detection. Low spring constant chips allow for a wide range of deflection however have a low threshold for force tolerance. High spring constant chips can detect high forces however have a narrow range of deflection. Three different chips representative of a low, middle, and high spring constant (5 N/m, 100 N/m, and 300 N/m) were tested through a deflection of 100 microns using a nanoindenter (Bruker, Minneapolis, MN, USA). A MEMS force probe on a manual stage (Femtotools, Buchs, Switzerland) was used to independently verify these results.

Blood Specimen Collection: All blood samples were collected according to the IRB protocol HSC-MS-19-0204 from 2 January 2020 to 3 January 2020 at the McGovern Medical School. Healthy male and female volunteers at least 18 years old were allowed to participate in the study. Participants included laboratory volunteers and medical students. Venipuncture was performed in the antecubital fossa with a 21 G needle into two 4.5 mL vacutainer tubes with 9:1 sodium citrate. Blood was kept at room temperature on a rocker and samples were used one hour after the blood draw.

Platelet Stimulation or Inhibition: Certain commercially available reagents cause either increased or decreased levels of pMLC through inhibition or activation of the regulatory pathways governing MLCK and MLCP (Figure 4). Sildenafil citrate is an inhibitor of phosphodiesterase-5 (PDE-5), which is an upstream inhibitor of the cyclic GMP governing activation of MLCP [33]. CPD7a is a quinolone derivative and similarly to sildenafil inhibits PDE-5 [34]. U46619 is a thromboxane analog and activates the extracellular signal-regulated kinase (ERK) that governs MLCK [35]. Phorbol 12-myristate 13-acetate (PMA) is a tumor promoter that directly activates protein kinase C which governs downstream ERK and CPI-17 [36]. Okadaic acid is a cell-permeable compound that directly inhibits MLCP [37]. ML-7 is a direct inhibitor of MLCK [23]. K252d is a indolocarbazole and acts in opposition of PMA to inhibit PKC [38]. Finally, Y-27632 inhibits the Rho kinase that normally inhibits MLCP, leading to decreased pMLC [39].

Platelet suspensions were aliquoted into 100 μL samples in Tris buffer, pH 7.4, containing the activators and inhibitors specified below. Tris buffer was used as it does not contain phosphates. The platelet stimulation was terminated by adding 400 µL of ice-cold Tris buffer and centrifuging the suspension at 1000× *g* for 15 min to collect the platelet pellet. Reagents were added to the platelet suspension at the following final concentrations with 20 min of incubation time: Sildenafil citrate (cat.#76021-842, VWR), 1.6 μM; U46619 (cat.# NC9787129, Fisher), 260 pM; PMA (cat.# P1585, Sigma), 300 pM; Okadaic acid (cat.# O7885, Sigma), 500 nM; Calcium chloride (cat.# AAL1319130, Fisher), 6 mM; CPD7a (cat.# 50-895-70001, Fisher), 1 μM; ML-7, hydrochloride (cat.# 475880, Sigma), 160 nM; K252d (cat.# 28469, Cayman Chemical), 16 nM; Y-27632, hydrochloride (cat.# 10005583, Cayman Chemical), 20 μM.

Measuring Clot Contraction: The activators and inhibitors in similar concentrations as mentioned above or saline control were added to whole blood samples and allowed to incubate at room temperature on a rocker for 15 min. Next 94 μL of the whole blood was mixed with six microliters of 100 mM CaCl2 and gently mixed. 35 μL of whole blood was immediately loaded onto the device by injecting the sample between two polymer plates and allowing it to coagulate.

Platelet Isolation: Platelets were isolated for testing in ELISA and Western Blotting. 1.5 mL of whole blood was aliquoted from each tube into 2 mL centrifuge tubes. Whole blood was spun at 200× *g* for 15 min the platelet-rich plasma (PRP) was transferred into empty 2 mL Eppendorf tubes and the red blood cells were discarded. PRP was centrifuged at 1000× *g* for 15 min to pellet the platelets. Platelet poor plasma (PPP) was discarded and the platelet pellet was left intact at the bottom of the tube. To wash platelets, 500 µL of ice-cold Tris buffer was added to re-suspend the platelet pellet (platelet resuspension was done by gently cycling the fluid using a 1000 µL pipette at 500 µL until the platelet pellet at the bottom of the tube is dissolved). The suspension was spun again at 1000× *g* for 15 min, the supernatant was discarded and platelets were resuspended in 1000 µL of Tris buffer. The platelets were kept on ice until ready to use in the assay. Platelet concentrations were determined by the volunteers’ natural platelet count.

ELISA: (Myosin regulatory light chain 2 (Phospho-Ser18) Colorimetric Cell-Based ELISA Kit, OKAG01692, Aviva Systems Biology). A 96-well ELISA plate was activated with poly-L-lysine (P4832, Sigma) for 30 min at 37 °C. The platelets were seeded in wells and incubated overnight at 37 °C. Platelet activators or inhibitors were added as described in the Platelet Modification section, directly on the ELISA plate. The plate was incubated at room temperature for 20 min and washed carefully so as not to disturb the platelet layer. The platelets were fixed and quenched according to the kit protocol. Then, 50 μL of MLC and pMLC rabbit anti-human primary antibodies were added and incubated overnight at 4 °C. Secondary goat anti-rabbit HRP-labeled antibodies were added for 1.5 h at room temperature on the shaker. TMB developer solution was added for 30 min and the reaction was stopped with the stop solution provided with the kit. The plate was read at OD 450 nm using VERSA Max Microplate Reader (San Jose, CA, USA) with 10 s auto mix before reading at room temperature. The data was analyzed using SoftMaxPro version 5.4 software (San Jose, CA, USA).

Western Blotting: Platelet activators and inhibitors were added as explained in the Platelet Modification section. Forty microliters of RIPA buffer were added to platelet pellets and kept on ice for 20 min and vortexed every 5 min to aid platelet lysis. The lysate was centrifuged at 13,000× *g* for 15 min to collect cell debris. The protein supernatant was kept and the pellet was discarded. Bradford assay was performed to account for protein concentration for equal loading into each well.

Western blotting was performed as previously described elsewhere. In short, proteins were separated on 12% Bis-Tris minigel in MES buffer system and electrotransferred onto a polyvinylidene difluoride (PVDF) membrane for immunoblotting. The membranes were blocked and incubated with the respective antibodies for MLC and pMLC (1:10,000, cat.# 3674 and #3672, Cell Signaling Technology (Danvers, MA, USA); β-actin loading control antibody (MA5-15738, ThermoFisher), 1:10,000 goat anti-mouse (cat.# 102971-068 LiCOR), and 1:10,000 H+L superclonal goat anti-rabbit (cat.#A27036, Fisher) and developed using chemiluminescent substrate (Licor). Image analysis was performed with Image Studio Digits Version 5.2.

Statistical Analysis: Statistical analysis was performed with Stata 14.2 (College Station, TX, USA). Force and phosphorylation data replicates are expressed as the median and interquartile range (IQR) in graphical form as box plots in Figure 4. In written form, this data is expressed as the mean and standard deviation.

## 5. Conclusions

Platelet contraction is governed by the kinase and phosphatase of the myosin light chain. A signaling cascade regulates levels of phosphorylated myosin light chain which drives the cycling of the myosin-actin subunits and ultimately contraction force during coagulation. Increases in phosphorylated myosin light chain correspond to increases in platelet contraction force. The relationship between contraction force and phosphorylated myosin light chain was proven using known activators or inhibitors of the kinase phosphatase pair. This study is important clinically because it identifies the myosin light chain as an attractive and reliable target for future drug treatments targeted towards platelet function.

## Figures and Tables

**Figure 1 ijms-22-06448-f001:**
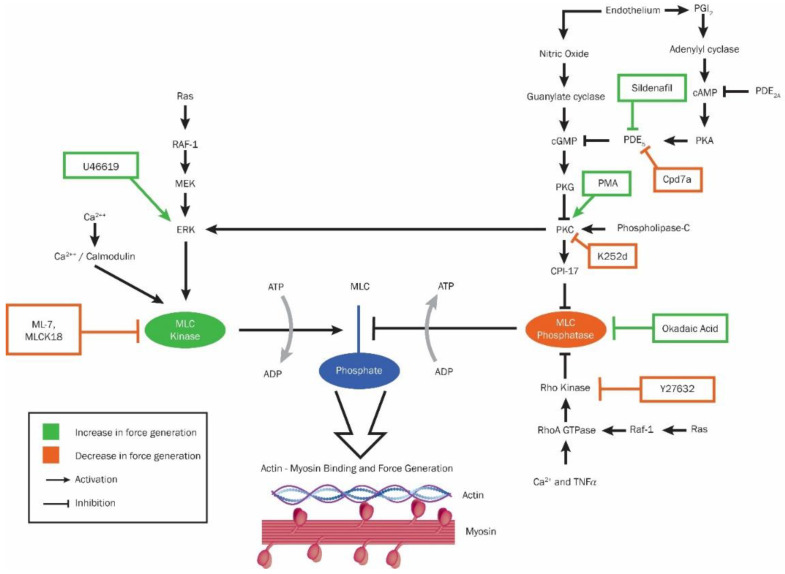
Actomyosin force generation diagram. Green boxes indicate reagents that lead to increased phosphorylation of myosin light chain (MLC) and increased platelet contraction force. Orange boxes indicate reagents that lead to decreased phosphorylation and decreased platelet contraction force. MLC kinase (MLCK) hydrolyzes ATP to phosphorylate MLC and MLC phosphatase (MLCP) does the opposite. Phosphorylation of MLC leads to the cycling of actin and myosin which creates contraction force. MLCK is activated by the calcium calmodulin complex and extracellular signal-regulated kinase (ERK) whose other upstream positive effectors include mitogen-activated protein kinase (MEK), the serine/threonine kinase RAF-1, and the small G-protein Ras. MLCP is controlled by a series of negative effectors. CPI-17 directly inhibits MLCP. CPI-17 is activated by protein kinase C (PKC) whose upstream positive effector includes phospholipase-C and negative effector includes cGMP-dependent protein kinase (PKG). PKG is controlled ultimately by milieu released by the endothelium—its positive effector nitric oxide and its negative effector prostacyclin (PGI_2_) that acts through the sequence of adenylyl cyclase, cyclic adenosine monophosphate (cAMP), protein kinase A (PKA), and phosphodiesterase type 5 (PDE_5_). PKC also feeds back to activated ERK and thus MLCK. The other MLCP negative effector is Rho-kinase which is controlled by the G protein RhoA. RhoA GTPase is activated by the kinase Raf-1, calcium, and tumor necrosis factor-alpha (TNFα).

**Figure 2 ijms-22-06448-f002:**
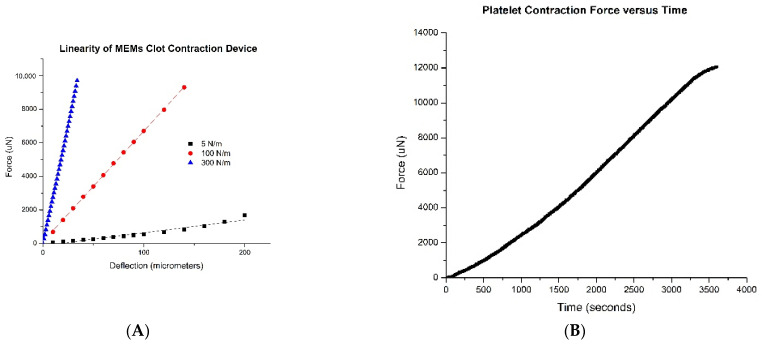
Verification of CCS deflection and force readings (**A**): A nanoindenter was used to deflect the sensor platform between 0 and 100 micrometers (the maximum deflection limit of the CCS) over a range of chip spring constants. A force probe attached to the nanoindenter recorded force production. Each CCS has its own predicted linear increase in force as deflection increases defined by Hooke’s Law, represented by the solid line. The discrete points near each line are actual force readings by the probe at a deflection recorded by the nanoindenter. Chips with a low spring constant will offer lower resistive force over a standard deflection range compared to chips with higher spring constants. *Raw Data from CCS* (**B**): The software system for the CCS calculates force from the recorded displacement of the tracking prism by the microscope objective. Force in micro Newtons is recorded versus time in seconds and a force curve is generated.

**Figure 3 ijms-22-06448-f003:**
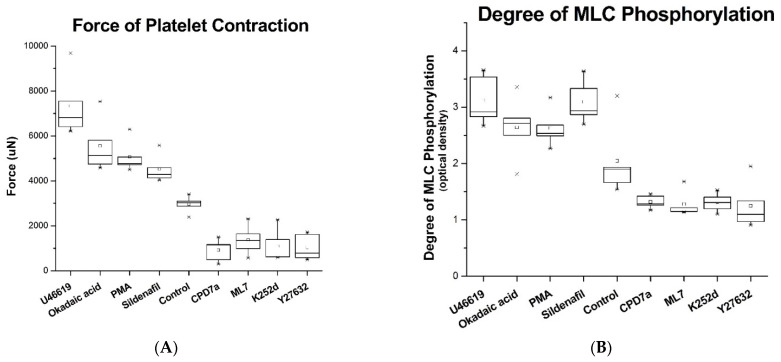
Force of platelet contraction (**A**) and degree of MLC phosphorylation (**B**) varied due to the addition of various stimulating or inhibiting compounds (n = 5, error bars = interquartile range). The control group was stimulated with only calcium. Stimulators like U46619, okadaic acid, PMA, and sildenafil demonstrate an increase in platelet contraction force and MLC phosphorylation. Inhibitors like CPD7a, ML7, K252d, and Y27632 cause a decrease in platelet contraction force and MLC phosphorylation. × represents outliers in the data set.

**Figure 4 ijms-22-06448-f004:**
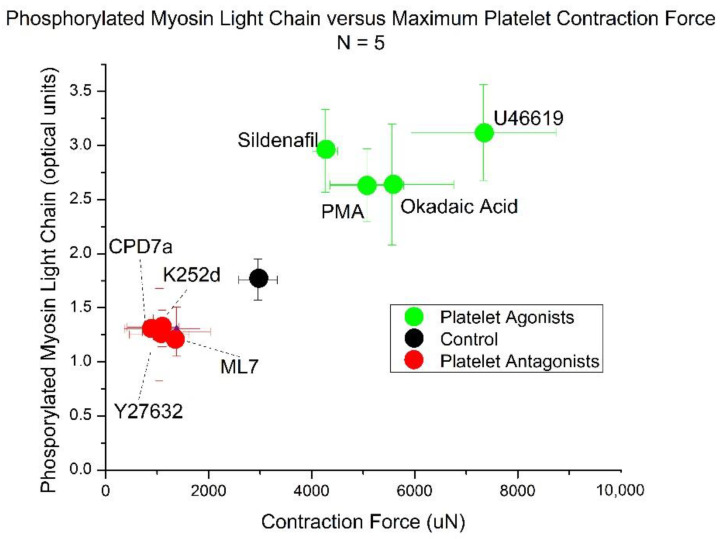
Phosphorylated myosin varies linearly with contraction force. Platelet antagonists shown in red cause decreased levels of phosphorylated myosin and corresponding decreases in platelet contraction force. Platelet agonists shown in green cause increased levels of phosphorylated myosin and increased contraction forces. Each point reflects an average of 5 different volunteers with standard deviation shown for contraction force and phosphorylated myosin. Contraction force varies linearly with phosphorylated MLC with an R squared value of 0.91.

**Figure 5 ijms-22-06448-f005:**
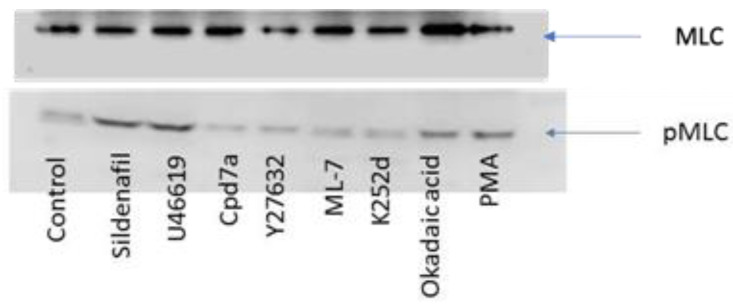
Western Blotting for pMLC detection. Platelet lysate was analyzed for the presence of pMLC after the addition of reagents. The addition of the activators U46619, sildenafil, PMA, okadaic acid resulted in increased MLC phosphorylation which coincided with increased platelet contraction forces. Blots are representative of five independent experiments. The total MLC band used to calibrate the increase in phosphorylation is included.

**Figure 6 ijms-22-06448-f006:**
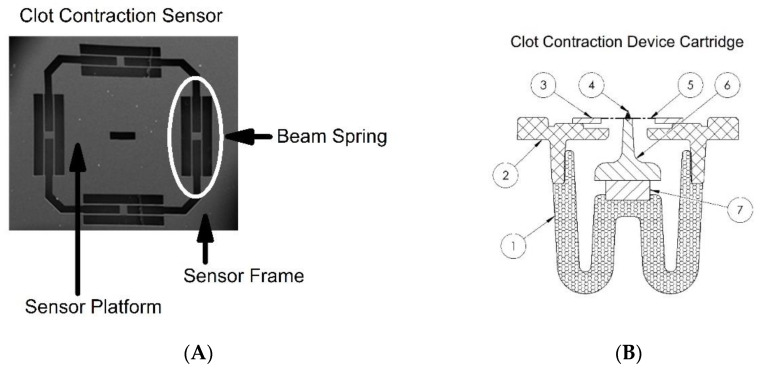
Scanning electron microscope image of the Clot Contraction Sensor (CCS) (**A**): The CCS consists of a sensor platform, symmetrically opposed beam springs, and a sensor frame. The beam springs allow linear travel of the sensor platform perpendicular to the sensor frame. The spring constant of the sensor is defined by the thickness and length of the beam springs. In the CCS shown, dimensions of the beam spring are 10 μm wide, 680 μm long, and 50 μm thick which defines a nominal spring constant of approximately 100 N/m. Platelet contraction induces translation of the sensor platform perpendicular to the plane of the sensor frame. The SEM image shown is approximately 4000 μm across. Clot Contraction Sensor and device cartridge in cross-sectional view (**B**): The various components of the CCS and device cartridge are labeled 1–7. Blood (7) is injected between two circular acrylic plates (1&6). One plate (1) is fixed; the other (6) is attached to the sensor platform of the CCS (5). The sensor frame (3) is fixed to the device platform (2). The sensor platform (5) is suspended from the sensor frame (3) using beam springs of known Hooke’s constant. A camera with a microscope objective focuses on a tracking prism (4) to determine the downward displacement of the sensor platform which yields contractile force by Hooke’s Law. Clot contraction device full assembly orthogonal view (**C**): The clot contraction sensor and device cartridge (1) sit within an aluminum housing (2). An LED (3) illuminates the tracking prism on the clot contraction sensor and a camera (4) with an attached lens (5) captures the prism location. All components are attached to a plastic base (6). Clot contraction device full assembly cross-sectional view (**D**): The clot contraction sensor with device cartridge (1) sit in the aluminum housing (2) and are shown in cross-section with the LED (3) out of view. The camera (4) and lens (5) are fixed to keep the tracking prism in a field of view. A heater (6) keeps the aluminum housing at physiologic temperature and is regulated by a thermocouple (7).

## Data Availability

Please email the corresponding author for data.

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
