# Peer review of "Microelectromechanical System Measurement of Platelet Contraction: Direct Interrogation of Myosin Light Chain Phosphorylation"

_ijms, 2021, doi:10.3390/ijms22126448_

Round 1

Reviewer 1 Report

The work by George et al. is solid and nicely presented.

I have a few questions that should be addressed for publication:

Of the drugs used in the study, the results with Sildenafil are surprising (even though MLC phosphorylation and contraction force suggest the same thing) because they are opposite to those obtained with CPd7a, another PDE inhibitor. In addition, the degree of MLC phosphorylation appears to be very high compared with the measured contraction force. It is quite possible that Sildenafil triggers other signaling pathways as explained in the results section and in the discussion (same sentence). But I am not sure that this unnecessarily complicates the message of the study.
The second point concerns the phosphorylation of MLC detected by western blotting, the authors should also provide image and measurement of total MLC to calibrate the increase in phosphorylation. Similarly, it is not clear how the elisa assay was calibrated; although the same number of platelets were seeded in the plate, how do the authors control the amount of platelets before the measurements? 

Author Response

The work by George et al. is solid and nicely presented. I have a few questions that should be addressed for publication:

Of the drugs used in the study, the results with Sildenafil are
surprising (even though MLC phosphorylation and contraction force
suggest the same thing) because they are opposite to those obtained with
CPd7a, another PDE inhibitor. In addition, the degree of MLC
phosphorylation appears to be very high compared with the measured
contraction force. It is quite possible that Sildenafil triggers other
signaling pathways as explained in the results section and in the
discussion (same sentence). But I am not sure that this unnecessarily
complicates the message of the study.   ​Thank you for this kind critique. We agree that our current discussion in the manuscript explains this issue. 
The second point concerns the phosphorylation of MLC detected by western
blotting, the authors should also provide image and measurement of total
MLC to calibrate the increase in phosphorylation.    Yes we regret not including the total MLC image, we have included the image used to calibrate the increase in phosphorylation here.
  Similarly, it is not clear how the elisa assay was calibrated; although the same number of
platelets were seeded in the plate, how do the authors control the
amount of platelets before the measurements?   The ELISA procedure was based on protocol provided by the Aviva Systems Biology. After platelets’ attachment to the surface, they were fixed, so we assume the same number of platelets that were successfully seeded, produced the signal. We have no control over how many attach to the surface, but out of 20,000 platelets treated with the same seeding conditions, I estimate similar number of platelets remain in each well. The activators and inhibitors were added after platelets attachment to the surface. MLC standard was used to calibrate the assays.

Reviewer 2 Report

The paper by George et al. is a very well-written paper, that describes in detail platelet contractility. I like this paper and it should definitively be published. This is a somekind review-and-data paper, that helps to unterstand the whole process of platelet contractile funktion. The authors acknowledge that many others have worked on this issue and that this paper contains confirmatory as well as new data. However, this little lack in novelty does not harm and should not prohibit publication. The authors provide two essential insights:

  1. Data on the Signal transduction pathway that regulates platelet contractile function
  2. They introduce and define a methodology to gain pretty exact measurements in platelet contractility   

However, there are a few points to address:

These points do need to be answered by further experiments, but they may be adressed in an extended discussion. 

  1. In your Fig 3 control platelets have a basal contraction force of around 3000uN. I do not  believe this. A control platelet should  be a quiescent platelet and its  contractile force should be zero, because a completely quiescent platelet does not contract. We know, that the platelet isolation process does activate the platelets to some degree, but they do not aggregate, because they are washed platelets in a fibrinogen-depleted experimental environment. With your inhibitors you are able to cool them somewhat down, but this process is in part irreversible. This is the reason, why you do not reach the "zero" force by inhibition.
  2. You prove this incomplete quiescence in Fig 3. MLC is slightly activated in the control. Although the authors state, that the inhibitors inhibit phosphorylation, the blot itself  does not really show this. The bands of cpd7a, ML7 and K252d are (almost) equal compared to control.  This is, because phpsphorylation in platelets is at least in part, if not completely, irreversible. This makes sense, because platelets that are caught in fibrin strands, are unable to move anyway.
  3. The points above are easily explained by the platelets isolation method: Blood is drawn in vacutainers. The vacuum und the shear force leads to platelet activation. Blood needs to be drawn through a big needle, that allows it to flow out without suction. Platelets are washed with ice-cold buffer.  This activated platelets as well, not the ice, but the seconds of rewarming activates the platelets immediately directly before the experiment starts.   Buffer at room temperature is much better. What is the reason for rocking (say: activating) the whole blood on a rocker? Tris buffer activates platelets as well, platelets should be kept at pH 7.4. Tris is not physiological.
  4. minor points:    p6 lines 167-170 and p7 lines 229-243: These are instructions for the authors. Delete these lines if you do not want to give the impression, that none of the authors has read the manuscript before submission.        Fig 6: put the (a) and (b) closer to the parts they belong to.  The (b) is right now closer to part (d) than to part (b).

Author Response

Thank you for this review. Our point-by-point responses to your comments are below:

1. In your Fig 3 control platelets have a basal contraction force of around 3000uN. I do not  believe this. A control platelet should  be a quiescent platelet and its  contractile force should be zero, because a completely quiescent platelet does not contract. We know, that the platelet isolation process does activate the platelets to some degree, but they do not aggregate, because they are washed platelets in a fibrinogen-depleted experimental environment. With your inhibitors you are able to cool them somewhat down, but this process is in part irreversible. This is the reason, why you do not reach the "zero" force by inhibition.

Thank you for this comment. To clarify, the basal contraction force shown in Figure 3 was from whole blood samples that were anticoagulated with citrate, and then re-calcified to initiate the clotting process. These samples will indeed demonstrate contraction and thus a detectable force. The platelet isolation process was only used for the ELISA and Western experiments. 

2. You prove this incomplete quiescence in Fig 3. MLC is slightly activated in the control. Although the authors state, that the inhibitors inhibit phosphorylation, the blot itself  does not really show this. The bands of cpd7a, ML7 and K252d are (almost) equal compared to control.  This is, because phpsphorylation in platelets is at least in part, if not completely, irreversible. This makes sense, because platelets that are caught in fibrin strands, are unable to move anyway.

Thank you for this comment. Your point regarding the irreversibilty of platelet phosphorylation is a very good one. This does explain why the degree of phosphorlyation change between controls and inhibitors is less than that between controls and activators. We have added discussion to this point on line 214 in the discussion section. 

3. The points above are easily explained by the platelets isolation method: Blood is drawn in vacutainers. The vacuum und the shear force leads to platelet activation. Blood needs to be drawn through a big needle, that allows it to flow out without suction. Platelets are washed with ice-cold buffer.  This activated platelets as well, not the ice, but the seconds of rewarming activates the platelets immediately directly before the experiment starts.   Buffer at room temperature is much better. What is the reason for rocking (say: activating) the whole blood on a rocker? Tris buffer activates platelets as well, platelets should be kept at pH 7.4. Tris is not physiological.

Thank you for the insightful comment. Regarding the use of Tris buffer, we used it because it is one of the few buffers that does not contain phosphates, thus it would not interfere with the ELISA and Western experiments. We have added this explanation to line 371 of the methods section. 

For our blood collection process - we used 21 gauge needles instead of larger 18 gauge needles since this is what was specified by our IRB protocol. And we uniformly place any collected blood sample on a gentle rocker (rocking 15 degrees every 5 seconds) to prevent stasis of the sample and potential aggregate or clot formation.

Regarding the temperature of the buffer - we will certainly take this into account for future experiments. 

4. minor points:    p6 lines 167-170 and p7 lines 229-243: These are instructions for the authors. Delete these lines if you do not want to give the impression, that none of the authors has read the manuscript before submission.        Fig 6: put the (a) and (b) closer to the parts they belong to.  The (b) is right now closer to part (d) than to part (b).

We have removed lines 167-170 and made spacing adjustments to improve Figure 6.